On the move: spatial ecology and habitat use of red fox in the Trans-Himalayan cold desert

Reshamwala Hussain S. 1
Raina Pankaj 2
Hussain Zehidul 1
Khan Shaheer 1
Dirzo Rodolfo 3
Habib Bilal bh@wii.gov.in 1
1 Wildlife Institute of India , Dehradun , Uttarakhand , India
2 Department of Wildlife Protection , Leh , Ladakh Union Territory , India
3 Stanford University , Stanford , United States of America
Kramer Donald
Electronic publication date: 2022 Sep 15
Publication date: 2022
Volume: 10
Electronic Location ID: e13967
Received 2021 Dec 23; Accepted 2022 Aug 8
Copyright: ©2022 Reshamwala et al.
Copyright year: 2022
Copyright holder: Reshamwala et al.
License: This is an open access article distributed under the terms of the Creative Commons Attribution License, which permits unrestricted use, distribution, reproduction and adaptation in any medium and for any purpose provided that it is properly attributed. For attribution, the original author(s), title, publication source (PeerJ) and either DOI or URL of the article must be cited.
License URL: https://creativecommons.org/licenses/by/4.0/

Keywords: Canidae, Daily movement, Home range, Ladakh, Space use, Step-selection function

Funding: Government of India’s National Mission on Himalayan Studies implemented by the Department of Wildlife Protection, Leh This research was supported by the project “Understanding Ladakh’s socio-ecological processes to design landscape level development strategies” funded by the Government of India’s National Mission on Himalayan Studies implemented by the Department of Wildlife Protection, Leh. The funders had no role in study design, data collection and analysis, decision to publish, or preparation of the manuscript.

==============================
Red fox (Vulpes vulpes) is the most widespread wild carnivore globally, occupying diverse habitats. The species is known for its adaptability to survive in dynamic anthropogenic landscapes. Despite being one of the most extensively studied carnivores, there is a dearth of information on red fox from the Trans-Himalayan region. We studied the home range sizes of red fox using the different estimation methods: minimum convex polygon (MCP), kernel density estimator (KDE), local convex hull (LoCoH) and Brownian-bridge movement model (BBMM). We analysed the daily movement and assessed the habitat selection with respect to topographic factors (ruggedness, elevation and slope), environmental factor (distance to water) and anthropogenic factors (distance to road and human settlements). We captured and GPS-collared six red fox individuals (three males and three females) from Chiktan and one female from Hemis National Park, Ladakh, India. The collars were programmed to record GPS fixes every 15-min. The average BBMM home range estimate (95% contour) was 22.40 ± 12.12 SD km2 (range 3.81–32.93 km2) and the average core area (50% contour) was 1.87 ± 0.86 SD km2 (range 0.55–2.69 km2). The estimated average daily movement of red fox was 17.76 ± 8.45 SD km/d (range 10.91–34.22 km/d). Red fox significantly selected lower elevations with less rugged terrain and were positively associated with water. This is the first study in the Trans-Himalayan landscape which aims to understand the daily movement of red fox at a fine temporal scale. Studying the movement and home range sizes helps understand the daily energetics and nutritional requirements of red fox. Movement information of a species is important for the prioritisation of areas for conservation and can aid in understanding ecosystem functioning and landscape management.

Introduction

One of the basic fundamentals for understanding an animal’s ecology is its home range size, which is defined as an individual’s area where movement occurs for normal activities like gathering food, mating and raising the young (Burt, 1943). The causes of these movements and their consequences are of great significance for understanding several aspects of an animal’s behaviour like dispersal (Small & Rusch, 1989), social interactions (Minta, 1993), space-use (Kenward et al., 2001) and population distributions (Turchin, 1991). Home ranges incorporate all individual’s movements and depend upon species behavioural and physiological responses to the environment (Horne et al., 2007; Vanak & Gompper, 2010). A mosaic of different habitat types is required to fulfil various life requisites, and these are reflected in the animal’s movement at larger spatial scales (Ims, 1995; Benson & Chamberlain, 2007). Intra- and inter-specific interactions in terms of competition, predation or facilitation also determine animal spatial distributions. Estimating the continuous movement of an animal is a powerful tool to quantify an animal’s daily movement and home-range size (Turchin, 1998). However, in nature, it is impossible to visually monitor animals continuously, especially in rugged terrain with adverse climatic conditions. The advancement of GPS telemetry is now commonly used to understand fine-scale movement and space use. It aids in continuous monitoring of animals over large distances and is particularly helpful for species that are elusive, nocturnal, or present in areas that are otherwise not accessible to researchers.

Red fox (Vulpes vulpes) is the most widespread wild carnivore in the world (Hoffmann & Sillero-Zubiri, 2016) and is known for its adaptability to survive in all sorts of environments (Geffen et al., 1996). The generalist and opportunistic nature of red fox in terms of their diet, habitat and movement patterns have contributed to its success (Geffen et al., 1996; Walton et al., 2017; Reshamwala et al., 2018). However, such mid-sized carnivores are also facing competition because of their broad diet, which overlaps with the energetic needs of myriad other sympatric competitors (Caro & Stoner, 2003). For example, a previous study reports a high diet-overlap of red fox with wolves and dogs from the Trans-Himalayan region (Reshamwala et al., 2021). As a result, they may have to make trade-offs either temporally or spatially (Schuette et al., 2013). Also, compared to the large carnivores, there have been fewer studies of meso-carnivores.

In this study, we first explore the home range sizes of red fox as estimated by different methods. We then estimate the average daily movement and assess the habitat use and selection in the arid Trans-Himalayan cold desert.

Material and Methods

Study area

The study was conducted in Ladakh, which belongs to the northwestern Trans-Himalayan region of India (Fig. 1). The area is characterised by dry rugged terrain with precipitation amounting to only 100 mm/year (Bharti et al., 2016). The cold desert has an elevation of 3000 to 7000 m above sea level and harsh winter temperatures, which go down to −30 °C. The vegetation is very sparse and patchy, with alpine meadows in certain areas (Kachroo & Dhar, 1977). The place is also sparsely populated with a human density of 4.9 individuals/km2 (Chandramouli, 2013). Foxes were GPS collared at two sites: Chiktan and Rumbuck. Chiktan is a small hamlet about 80 km from the city of Kargil. The people belong to the Muslim community and due to higher non-vegetarian food consumption, there is more anthropogenic food subsidy and greater fox abundance in this area as compared to other parts of Ladakh (Reshamwala et al., 2018; Reshamwala et al., 2021). Rumbuck is a village with few scattered human settlements within Hemis National Park, and the people belong to the Buddhist community. People are primarily associated with agriculture or agro-pastoralism and are sometimes involved with tourism, especially in Hemis National Park (Reshamwala et al., 2021). Besides red fox, snow leopard (Panthera uncia), wolf (Canis lupus), Ladakh urial (Ovis vignei), blue sheep (Pseudois nayaur), ibex (Capra ibex), weasel (Mustela altaica), and stone marten (Martes foina) occur in this area.

Figure 1 Study area of collared red foxes in the Trans-Himalayan cold desert, Ladakh, India.

(A) Location of the study sites in India. (B) 95% minimum convex polygon (MCP) home ranges of six red foxes at Chiktan, Kargil. (C) 95% MCP home range of a female at Hemis National Park.

Fox capture and GPS-collaring

We captured seven adult red foxes from October 2018 to January 2019 using five Victor soft catch #3 leg hold traps. Six individuals were captured in Chiktan and one in Hemis National Park. The animals were immediately collared with Sirtrack collars (Litetrack 150 iridium, weighing ∼200 g) and released within 20–30 min at the place of capture. Animal handling, capture and release were approved by the Department of Wildlife Protection, Jammu and Kashmir (CCFWL\Permission\2016\575-76). No drugs were administered to the animals, and the methodology was refined to ensure minimum stress, handling time, and injury to the captured individual, approved by Wildlife Institute of India, animal ethics committee. The foxes were weighed and sexed before release (Table 1). Weight was determined by weighing a researcher holding the fox and then subtracting the researcher’s weight. The collars were programmed to record GPS fixes at 15-min intervals and transmit the same at 3-h intervals. We chose a 15-min interval as previous studies recommend this time for calculating the daily distance travelled by animals (Musiani, Okarma & Jȩdrzejewski, 1998). The collars had an auto drop-off scheduled for 364 days and none of the foxes died during the study duration. Due to the 15-min fixes the battery lasted from 56 to 90 days for five individuals. The GPS fixes from F2 were most irregular and hence the battery lasted only 56 days. For individuals M3 and M4, the collars were reprogrammed to take GPS fixes at 130-min intervals after one month to increase the battery life. These collars gave data for 198 and 212 days, respectively. At the time of collaring, none of the individuals had pups. Later while tracking, F1 and F2 were sighted with pups in the month of April. The collaring process was conducted in winter, which coincides with the breeding time of foxes in Trans-Himalayas.

Table 1 Details of GPS-collared individuals, date of collaring, active days, and the number of locations received to study the spatial ecology of red fox in Ladakh, India.

ID	Sex	Weight (kg)	Date of collaring	No. of days collared	Total GPS fixes	
F1	F	6.0	27/10/2018	90	6,820	
F2	F	5.6	17/12/2018	56	3,613	
F3	F	5.2	24/12/2018	79	3,208	
M1	M	7.1	29/10/2018	83	5,120	
M2	M	8.2	17/11/2018	75	4,425	
M3	M	6.2	19/01/2019	198	4,235	
M4	M	6.4	22/01/2019	212	3,840	

Data analysis

The data from seven red foxes included 31,261 GPS fixes and were filtered in ArcGIS 9.2.1 with the ArcMET filter tools (Wall, 2014). The filtered data consisted of 28,163 GPS fixes and excluded fox locations that exceeded 48 km/h as this is the highest speed of the red fox (Haltenorth & Roth, 1968). Home ranges were estimated using minimum convex polygon (MCP), kernel density estimator (KDE), local convex hull (LoCoH) and Brownian-bridge movement model (BBMM) from the ArcMET utilisation distribution and range tools (Wall, 2014). We used MCP and KDE methods as they are used widely to estimate home range size of red fox, allowing comparison with other studies. Rugged terrain often force animals to follow a fixed path. In such areas, LoCoH gives a reliable home range for animals as this method can identify rigid boundaries such as rivers, lakes or otherwise inhospitable terrain (Getz et al., 2007). The BBMM is the most recent method which incorporates the time and probability of movement; it thereby models the animal movements and helps determine animal home ranges more robustly (Horne et al., 2007). Since each method has its own advantages, we estimated home ranges using all of these methods. We used each of these different methods to evaluate the home ranges (95% contour) and core areas (50% contour) of each individual. We also calculated the net square displacement from our collaring location using the path tool details to identify dispersing individuals.

We calculated daily movements, i.e., the sum of displacement in one day, with the help of trajectory details from the ArcMET path statistics tool in ArcGIS. We considered the daily movement as the sum of linear distances obtained from each consecutive GPS fix from the first location of midnight and the last location of the same day. For M3 and M4 individuals, we use only the 15-min data for estimating the daily movement. While estimating the daily movements, we found an abrupt change in trajectories in certain instances. On inspecting the spatial data further, we could identify irregular spikes in the speed of these trajectories. Hence to remove these high movements, we truncated data above 98% (Fig. S1). It is possible that some of these data were actual movements, but to avoid errors while estimating daily movement, we have removed them from the analysis.

Since GPS fixes obtained from larger time intervals can affect the estimation of daily movement of animals, we also down-sampled our data to estimate the reduction in the movement. Most GPS telemetry studies have GPS fixes scheduled for time intervals of 1-h or greater for longer battery life. However, the daily average movement may be greatly under-estimated at this time interval (Poulin, Clermont & Berteaux, 2021). Hence, to evaluate this effect of GPS fixes, we resampled our 15-min GPS fixes data at 30, 60 and 120-min time intervals and calculated the daily average movement. We used one-sample t-test to evaluate the differences.

To understand habitat use, we assessed the percentage of steps in each land cover category for every individual. We used the Biodiversity Information System portal of the Indian Institute of Remote Sensing (https://bis.iirs.gov.in) for land cover data which consisted of nine classes (Roy et al., 2015). We then re-classified land cover data into agriculture, vegetation, barren and snow. Agriculture consisted of cultivated land, whereas vegetation consisted of moist alpine scrub, dry alpine scrub, moist alpine pasture and dry alpine pasture. Barren land consists of no vegetation and snow refers to areas covered with permanent snow.

For habitat selection in relation to topography (ruggedness, elevation, and slope), an environmental factor (distance to water) and anthropogenic disturbances (distance to road and to human settlements), we used integrated step-selection functions (iSSF) from the amt package in R (Signer, Fieberg & Avgar, 2019). iSSF jointly estimates habitat selection and animal movement parameters (e.g., step length and turn angle) by relaxing the implicit assumption that these are independent (Signer, Fieberg & Avgar, 2019). This method assesses habitat preference in animals by comparing each used step (i.e., movement between two consecutive GPS fixes) to those of randomly placed steps (i.e., that animal could have taken) within the movement path. The random steps are generated using distances sampled from a gamma distribution fitted to the empirical step length distribution and random turning angles by von Misses distribution. We generated both true and random steps for 15-min intervals for all the individuals. We produced 10 random steps per used step, based on the recommendations of Thurfjell, Ciuti & Boyce (2014). At the end of each step, we extracted environmental covariates. All variables were scaled, centred and screened for collinearity using Pearson’s correlation coefficient with a threshold of —r—>0.7. We performed an iSSF using conditional logistic regression for the variables: distance to road, distance to water, distance to human settlements, ruggedness, elevation, and slope. We performed the step selection modelling at an individual level as it is preferred over the population level (Thurfjell, Ciuti & Boyce, 2014). Further, our sample size of seven individuals also advocates for individual modelling.

Results

Home range

The red foxes we observed showed high variation in the size of their home ranges and core areas (Table 2). We also observed overlapping home ranges at Chiktan. F3 showed large displacement from its initial place of collaring (Fig. S2), suggesting that it was either nomadic or dispersing. Similar displacements were not shown by any other individual. Because it is important not to include nomadic foxes while calculating home ranges (Meia & Weber, 1995), we excluded F3 from our home range calculations. The average BBMM home range (95% contour) was 22.40 ± 12.12 SD km2 and the average core area (50% contour) was 1.87 ± 0.86 SD km2. The smallest home range was for individual F1 (3.81 km2) whilst M1, M2, and F3 utilised multiple core areas (Fig. S3). The average female home range was smaller (16.78 km2, BBMM) than that of males (25.22 km2, BBMM). However, the difference between sexes was not statistically significant, probably due to the limited sample size and high variability among individuals. In our study, the average LoCoH and BBMM methods had a smaller estimated home ranges (LoCoH 17.33 ± 15.29 SD km2 and BBMM 22.40 ± 12.12 SD km2) than MCP method (MCP 43.24 ± 40.52 SD km2).

Table 2 Estimated home range sizes by minimum convex polygon (MCP), kernel density estimate (KDE), local convex hull (LoCoH), Brownian-bridge movement model (BBMM), and estimated average daily movement of red fox in Ladakh, India.

Individual	MCP (km2)	Kernel (km2)	LoCoH (km2)	BBMM (km2)	Daily average movement (km/d (SD))	
	Core area (50%)	Home Range (95%)	Core area (50%)	Home Range (95%)	Core area (50%)	Home Range (95%)	Core area (50%)	Home Range (95%)		
F1	0.14	1.95	0.22	1.62	0.05	1.17	0.55	3.81	11.10 (7.72)	
F2	0.36	45.99	1.27	41.59	0.32	37.57	2.33	29.75	20.92 (15.12)	
M1	1.74	118.77	1.50	18.49	0.18	8.97	1.93	31.02	11.61 (10.38)	
M2	24.47	39.36	4.02	26.65	0.26	10.63	2.62	26.19	10.91 (7.69)	
M3	1.01	37.50	1.57	38.83	0.43	35.57	2.69	32.93	34.22 (20.85)	
M4	0.51	15.90	0.77	12.07	0.36	10.11	1.15	10.75	20.85 (13.88)	
F3	–	–	–	–	–	–	–	–	14.72 (9.10)	
Average (SD)	4.70 (9.69)	43.24 (40.52)	1.55 (1.30)	23.20 (15.53)	0.26 (0.13)	17.33 (15.29)	1.87 (0.86)	22.40 (12.12)	17.76 (8.45)	

Daily movement

The estimated average daily movement of red fox was 17.76 ± 8.45 SD km/d (Table 2). The daily average movements of males were larger (19.39 ± 10.87 km/d, n = 4) than those of females (15.58 ± 4.96 km/d, n = 3). Individual M3 showed the highest variation and average daily movement (34.22 ± 20.85km/d). The daily movement of our largest individual M2, was least (10.91 ± 7.69 SD km/d, Fig. 2). The downscaling of data from 15-min GPS fixes to 2-h interval fixes significantly reduced the estimated daily average movements (P < 0.05, Table S1).

Figure 2 Violin plots showing the estimated daily movement of red fox in the Trans-Himalayan cold desert, Ladakh, India.

Values below the violin indicate the number of days used for estimating the average daily movement. Each dot in the violin plot represents the distance walked in a day. (Females = F1, F2, F3; Males = M1, M2, M3, M4).

Habitat use and selection

The barren land is most prevalent in the Trans-Himalayan cold desert and all foxes exploited this land cover category. The individual F1 was found more in agricultural land (83.65%) than other land-use categories, while F2 was mainly found in barren land (70.14%). M1 and M2 were present in agriculture and barren land. Similarly, M3 and M4 exploited agriculture and barren land, but also used areas with snow in higher proportions (50.47% and 63.71%, respectively). F3 vixen in Hemis National Park used only barren land (62.32%) and alpine vegetation (37.67%), respectively (Fig. 3).

For habitat selection, we found the red fox selected lower elevations and less rugged terrain (P < 0.001, Fig. 4). With respect to human settlements, only F1 showed significant negative selection (P < 0.001), while F3, M2, M3 and M4 selected to be near human settlements (P < 0.01). Individuals F2 and M1 did not exhibit significant selection for or avoidance of human settlements. The red fox tended to select for water bodies. However, for M3 and M4 individuals, this trend was non-significant. Except for F2 and F1, all individuals avoided roads and slope (P < 0.001).

Discussion

Home range

We found high variability not only across different methods for home range estimation, but also amongst individuals. Across different methods for home range estimation, the highest variation was found for the individual M1 whose home range varied from 118.77 km2 (95% MCP) to 8.97 km2 (95% LoCoH). We also found the largest home range for M1 (118.77 km2 95% MCP) while the largest core area was found for M2 (24.47 km2 50% MCP). The MCP method is adversely affected by extreme locations resulting in larger home range estimates. However, the core area of M2, the largest individual, overlapped with the core areas of all other individuals. The traditional methods MCP and kernels utilisation distribution allow comparisons between studies but may overpredict the actual home ranges because of exploratory behaviour. Since our study site consists of rugged terrain and we used short interval between GPS fixes, LoCoH and BBMM are likely to be better methods for home range estimation. The LoCoH method identifies steep slopes and inaccessible areas while BBMM incorporates space, time and animal-specific parameters or habitat (Walter, Onorato & Fischer, 2015), which accounts for highly autocorrelated movement locations.

Figure 3 Percentage of land cover categories utilised by different red fox individuals in the Trans-Himalayan cold desert, Ladakh, India.

Figure 4 Step-selection function beta coefficients* of different red fox individuals for topographic factors (ruggedness, elevation and slope), an environmental factor (distance to water) and anthropogenic factors (distance to road and human settlements).

*Beta coefficients derived from step selection function analysis. Positive values show preference and negative values show avoidance.

Our collared individuals had extensively overlapping home ranges at 95% MCP (Fig. 1). The high amount of overlap could be related to the small area in which trapping was conducted at Chiktan. However, the core areas had very little to no spatial overlap (50% MCP). The individual F1 consistently had the smallest estimated home range across different methods as compared to the other individuals. F1 and M2 were sighted together in several instances. Unfortunately, the den of this pair was inaccessible due to the dense seabuckthorn (Hippophae spp.) shrub. Additionally, due to the high anthropogenic food subsidy prevalent at the study site of Chiktan (Reshamwala et al., 2018; Reshamwala et al., 2021), the foxes may have a high tolerance and overlapping home ranges (Newdick, 1983).

The home range estimates using the BBMM method varied from 3.81 to 32.93 km2. The home range sizes of red fox are known to have significant variations ranging from as small as 0.40 km2 in urban areas of Oxford to as large as 40 km2 (MCP) in the Arctic or even more extensive which are determined majorly by the type of habitat (Sillero-Zubiri, Hoffmann & Macdonald, 2004). Such results are not uncommon and the home ranges of fox within the same area may often show wide variation. Similarly, in Illinois, U.S.A, the home range of red fox varied from less than 1 to >35 km2 (Gosselink et al., 2003).

Our study site is a high altitude cold desert with the presence of anthropogenic food subsidies. There have been no studies pertaining to the home range sizes of red fox from the Trans-Himalayan region. However, at higher latitudes and altitudes, animals tend to have larger home ranges (Mattisson et al., 2013; Morellet et al., 2013). A recent study on red foxes living in the Arctic reported home ranges as large as 60–72 km2 (Lai et al., 2022). Foxes living at higher elevations are reported to have four times more extensive home ranges than at lower elevations (Walton et al., 2017). In case of deserts, larger home ranges of red fox were reported in the arid Simpson Desert (Newsome, Spencer & Dickman, 2017). In contrast, the use of smaller home ranges by medium-sized canids near human settlements has also been reported (Coman, Robinson & Beaumont, 1991; Saeki, Johnson & Macdonald, 2007; Rotem et al., 2011). In areas with high human impact and food availability in abundance, the home ranges of red fox tends to be smaller (Main et al., 2020; Newdick, 1983). On one hand, the arid Trans-Himalayan cold desert with high elevation suggests larger home ranges. On the other hand, the presence of anthropogenic food subsidies may result in a smaller home range. In the case the foxes we studied, the two factors could together explain the observed high variation in home range sizes.

Daily movement

The estimated average daily movement of red fox in our study was 17.76 ± 8.45 SD km/d and ranged between 10.91 km/d and 34.22 km/d (Table 2). A study has reported the average daily movement of red fox, which was 4.8 to 16 km/d with a mean of 9.4 ± 3.7 km/d (Carter, Luck & McDonald, 2012). Another study has also reported similar daily movements of 3.9–12 km/day (Meia & Weber, 1995). However, the very high frequency (VHF) telemetry used in these studies has limitations such as continuous monitoring, inaccessible areas, monitoring multiple individuals, following nocturnal animals, etc. The average daily movement travelled by arctic fox (Vulpes lagopus) has been comparatively well studied and is reported to be 51.9 ± 11.7 km/d (Poulin, Clermont & Berteaux, 2021). The average daily movement of wolves showed a great variation, ranging from 17 to 38 km/d (Ciucci et al., 1997). The daily average movement of wolves travelled during the mating season may go as high as 34 km/d (Jedrzejewski et al., 2001). During the breeding season, male lynx frequently move longer distances and at greater speeds to increase their chances of mating (Schmidt, 1999). Since our data collection period coincides with the breeding time of red fox, we suspect M3 and M4 to have made similar extended forays (Fig. 2).

We found that our estimated daily average movement data were positively skewed. On most occasions, we observed small daily average distances, but there were a few instances of very high movements. The individual M3 showed the highest daily movement of 81.98 km and 79.64 km on two occasions. F2 also showed two events with high movements of 67.61 and 53.54 km. Foxes with smaller home ranges may not necessarily travel less. F1 had a very small home range (3.81 km2, BBMM) as compared to other individuals, but its average daily movement was 11.10 ± 7.72 SD km/d. Daily distance travelled by individuals having a smaller home range may be similar to individuals having a larger home range, and the difference may only reflect the different foraging sites used by the individuals (Carter, Luck & McDonald, 2012).

It is essential to have the GPS fixes at fine-scale time intervals to calculate the average daily distance travelled by red fox. Our data suggest that increasing the time interval of GPS fixes decreases the estimate of daily movement of red fox significantly (P < 0.01, Table S1). The estimated daily average movement of foxes decreased from 17.76 ± 8.45 SD km/d to 14.96 ± 7.71 SD km/d on down sampling the data from 15-min to 30-min time intervals. Further down sampling resulted in greater under-estimates of daily average movement (Table S1). On the other hand, fixes at short time intervals may overestimate the average daily distance travelled due to location errors (Poulin, Clermont & Berteaux, 2021).

The individual M3 showed a higher daily average movement (34.22 ± 20.34 SD km/d, Table S1). This may be because M3 made many forays outside its normal home range during this period. On four occasions, M3 moved about 60 km or more in one day. Such long distances are not uncommon, and in arctic fox, the maximum distance travelled in a day was reported to be 154 km (Fuglei & Tarroux, 2019). Long-distance dispersal events are similarly reported in red fox, where foxes have travelled over a distance of 132–1036 km in a short period (Walton et al., 2018).

Data on average speeds travelled by red fox have not been reported, but the average speed with which wolves travel ranges from 0.8 to 1.1 km/h (Burkholder, 1959; Ciucci et al., 1997; Jedrzejewski et al., 2001). Studies on wolves with 15-min GPS location data have reported maximum travel speeds ranging from 9.6 to 13 km/h (Mech, 1994). Hence, our method of restricting the maximum speed limit of 9.35 km/h for red fox may be reasonable. The use of speed to identify GPS errors, especially for calculating daily movements, has been reported in other studies (Bjørneraas et al., 2010; Wysong et al., 2020). Studies with a greater sample size are warranted to enhance our knowledge of maximum speed of red fox.

Habitat use and selection

Multiple land cover categories were exploited by different individuals in our study. Most of the area at our study site is barren due to the arid Trans-Himalayan conditions. The high elevation areas (>6000 m above sea level) are covered with snow all throughout the year. Agricultural land is scarce and present near the river valleys. We found that the larger foxes, i.e., F1, M1 and M2, exploited agricultural land the most as compared to any other land cover category (Fig. 3). On the other hand, F2, M3 and M4, which were smaller in size were found in barren and snow-covered land (Table 1, Fig. 3). We suspect that the larger individuals are more dominant, giving them greater access to preferred, resource-rich areas such as agricultural land. However, the overlapping home ranges indicate that the larger individuals do not completely exclude others from their home ranges. The individual F3 in Hemis National Park was found in both barren and alpine vegetation areas.

Red foxes are known for their highly adaptable behaviour and often show large variations in their habitat selection (Walton et al., 2017; Walton, 2020). In our study, except for F1 and F2, all other individuals were positively associated with human settlements. Red fox in the Trans-Himalayan landscape has been reported to be at higher densities and often den near human settlements due to the presence of anthropogenic food subsidies (Reshamwala et al., 2018; Reshamwala et al., 2021). We found that all individuals were positively associated with water. The presence of water bodies is also known to positively influence the abundance of rodents and lagomorphs (Reshamwala et al., 2021). We also found that red fox significantly preferred lower elevations and less rugged terrain. Except for F1 and F2, all other individuals significantly avoided slopes. Other species have been shown to avoid higher elevations, steep slopes and rugged terrain, which are known to be energetically costly to use (Filla et al., 2017; Fullman, Joly & Ackerman, 2017). At the time they were radio-collared, these individuals were without pups, but later field observations showed that both had young ones. They may have avoided human settlements and showed more preference for slopes to avoid anthropogenic disturbances.

Studies on the red fox population in the Trans-Himalayan region are limited and their current status remains unknown. However, a previous study has reported few retaliatory killings due to predation on poultry (Reshamwala et al., 2018). This study provides an overview of the spatial ecology of red fox in the Trans-Himalayan cold desert. The role of spatial ecology in conservation is well known and information on species movement is important for the prioritisation of areas for conservation (Allen & Singh, 2016; Carwardine et al., 2012). While the movement within the home ranges is crucial for understanding the habitat suitability and preferences (Lu et al., 2012), the home range sizes and shapes are essential for managers to understand the scale of management (Schwartz, 1999). From our movement data of GPS-collared individuals, we conclude that they preferred lesser rugged terrain and lower elevations in this region. In addition to home ranges, we provide insights on the daily movements and the need for having short time intervals between locations for calculating daily movements. The movement of species also influences other ecosystem services such as pollination and seed dispersal (Kremen et al., 2007; Nathan & Muller-landau, 2000). Hence, movement ecology can aid in understanding ecosystem functioning and landscape management (Habib et al., 2021; Mitchell, Bennett & Gonzalez, 2013).

Supplemental Information

Supplemental Information 1 Density distribution of speed data (values are log-transformed) from trajectory details of all red fox individuals and the cut-off line in red at 98% (9.35 km/h) of the data to eliminate movement errors

Click here for additional data file.

Supplemental Information 2 Net square displacement (km2) of collared individuals, 3 females (F1, F2, F3) and 4 males (M1, M2, M3, M4) in Ladakh, India. Dashed lines show the trendline for each individual

Click here for additional data file.

Supplemental Information 3 Brownian-bridge movement model (BBMM) utilisation distribution of home ranges (95% contour; black outline) and core areas (50% contour; red) of red foxes obtained from home range analysis done in ArcGIS. *Map scales for each individual are different

Click here for additional data file.

Supplemental Information 4 Effect of down-sampling GPS fixes from 15-min to 2-h time interval on the estimated daily movement of red fox and t-test results

Click here for additional data file.

Supplemental Information 5 Data from the MahaData Portal

Click here for additional data file.

We are thankful to the Director and Dean, Wildlife Institute of India, for support and encouragement. We are grateful to the Department of Wildlife Protection, Govt. Jammu and Kashmir for providing the necessary permissions. Gianalberto Losapio is acknowledged for his valuable inputs on the manuscript. Mr. Lavpreet Singh Lahoria, Mr. Hasnain Zargar and Mr. Ali Akbar Zargar are acknowledged for their help in capturing the foxes. Dr. Ajaz Hussain and his family are acknowledged for their constant support during this long-term study. Ms. Keerthi Sudarsan and Dr. Ranjana Pal are acknowledged for helping in editing the manuscript. The authors acknowledge a grant from Idea Wild for providing the necessary field equipment.

Additional Information and Declarations

Competing Interests

Author Contributions

Animal Ethics

Data Availability

The authors declare there are no competing interests.

Hussain S. Reshamwala conceived and designed the experiments, performed the experiments, analyzed the data, prepared figures and/or tables, authored or reviewed drafts of the article, and approved the final draft.

Pankaj Raina performed the experiments, analyzed the data, authored or reviewed drafts of the article, and approved the final draft.

Zehidul Hussain analyzed the data, prepared figures and/or tables, authored or reviewed drafts of the article, and approved the final draft.

Shaheer Khan analyzed the data, prepared figures and/or tables, authored or reviewed drafts of the article, and approved the final draft.

Rodolfo Dirzo analyzed the data, authored or reviewed drafts of the article, and approved the final draft.

Bilal Habib conceived and designed the experiments, authored or reviewed drafts of the article, and approved the final draft.

The following information was supplied relating to ethical approvals (i.e., approving body and any reference numbers):

All red foxes were captured following standard and approved protocols after due permission from the Ministry of Environment, Forests and Climate Change, Government of India, and Department of Wildlife Protection, Ladakh, Jammu and Kashmir. The permit details are as follows: CCFWL Permission 2016 575-76.

The following information was supplied regarding data availability:

The data is available at the MahaData Portal:

https://mahadata.wii.gov.in/documentreference/index.html.

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
