# Peer review of "On the move: spatial ecology and habitat use of red fox in the Trans-Himalayan cold desert"

_PeerJ, doi:10.7717/peerj.13967_

## Round 0.1 · original submission · Major Revisions

Overview
Both reviewers consider that this is a useful study of a red fox population and habitat for which little information is available. However, both indicate that the manuscript as submitted is far from ready for publication. There are many errors, inconsistencies, and information gaps, and it does not meet international standards for presentation, including organization, writing style and clarity of figures and tables.

You have not provided access to the raw data as required.

Reviewer 2 indicated that they were unable to find the supplementary table and figures. However, Reviewer 1 and I could see them. I will ask PeerJ staff to check whether there was a problem in the information sent to the reviewer.

I have provided an annotated pdf with highlights indicating problems I noticed. On some of these, my inserted comments identified the problem. Others are errors in grammar or word use that can be determined with the help of a reader more familiar with English grammar. This is by no means a complete list, so the help of one or more readers of the revised manuscript is essential. My annotated pdf started with the pdf provided by Reviewer 2. You can distinguish our comments because those of the review are identified by ‘Anon’ whereas mine use my initials (DLK). The inserted comments do not need to be included in your response if you agree and make suggested changes. If you do not agree, you should add them to your rebuttal document.

Editor’s Comments
General issues
• Grammatical and word use errors are too numerous to completely document. Commas are missing, article use is frequently incorrect, singular and plural use is inconsistent. The reviewers and I have indicated some of these, but there are many more. Before resubmitting the manuscript, please ask one or more readers familiar with scientific communication and English grammar to carefully read the manuscript. It is unreasonable to expect reviewers and editors to detect and correct so many errors in language use.
• The current text is quite wordy with many unnecessary redundancies. Try to revise the manuscript in a clearer and more concise form. (For example, you do not need to keep mentioning ‘in the red fox’ when the context is unambiguous.)
• Check that all unit abbreviations conform to SI (kms, hr are not standard abbreviations in the International System of Units).
• Either U.S. or U.K. spelling is acceptable, but a mix as in your manuscript is not acceptable. Use a spell checker to be sure that all use conforms to the same system, include in tables and figures (except that references use the original spelling, of course).
• Either indent paragraphs or skip a line between them so that the paragraph structure is clearly visible.

Abstract
L32-33. Should you mention the evaluation of different measures of home range or at least specify what measure these values were based on? Why only range and not measures of central tendency and variability (mean or median and SD or CI, depending on whether data correspond to parametric assumptions).
L35. Is the effect of interval on estimated movement important? It seems obvious to me that this would be the case, and I presume that this is well established. Also, the interval does not reduce the movement, but only the estimated movement.
L37 and elsewhere. Why do you regard p>0.01 as non-significant? (You don’t need p-values in the abstract, but this use occurs elsewhere in the manuscript.)
L37-43. These statements are too generic and could have been written before the study. What did your study actually contribute to our understanding of fox space use and movement in this area compared to other areas.

Introduction
The Introduction needs a much stronger focus. Try to be clear and specific rather than providing simple or vague generic statements. For example the first sentence states that space use is influenced by behavioral responses to the environment. The second sentence states that it is also influenced by ecological factors. Why you would make a distinction between response to the environment and ecological factors is unclear. Why you would introduce physiological responses as a separate category as if it does not involve behavior is unclear when the manuscript is about behavior.

For each of your measures, you can have a paragraph indicating its importance to ecology and behavior and perhaps conservation. The paragraph would then provide a clear, logical development identifying the relevant background and the knowledge gap that your study will attempt to fill. The final paragraph should explicitly list your objectives in the same order as will occur in the Methods and Results.

It is not clear to me how you could expect consistent directionality in a long-term home range study. If the fox moves in one direction, it has to move back. It can’t continue to move in the same direction if it has a stable home range.

Methods
Note that Methods must be clear and complete enough for another researcher to duplicate the work.
Specify the recording period of each individual (in a table) and note (in the text) how the season of study relates to the phenology of the population (e.g. mating season, snow and temperature patterns)
L136. Details of weighing and measuring methods needed.
L148-150. It is not clear what displacement and angular data refer to. Their position implies it is something about home range, but I don’t see how they are related.
L152. This paragraph refers to errors but fails to explain how movement was actually calculated. Also, what were the starting and ending points of a day? It could make a difference in the variation between days if not to total distance if you consider a day to start and end at midnight, noon, or some other time.
L159. I don’t think that the habitat selection methods are sufficient for another researcher to repeat the work. Were the random steps variation in the angle alone or angle and distance? What were the criteria by which the actual vs. random steps were judged? How did you assess whether any difference was statistically significant? The variables are not defined, and the source for the information is not specified. What was the purpose of the land cover map? Distinguishing vegetation and barren might be a challenge because it is probably a continuum.
L165. Why is there no method for directional preference or downscaling analysis, both referred to in the Abstract?

Results
• The Results do not follow the same order as Methods.
• For home range sizes, why do you only present maximum and minimum? Variation is important, but I would think that central tendency and variability are also relevant. Note that you should not use parametric measures of variability if the data are not normally distributed.
• Do you think that a map showing the home ranges of each individual, perhaps in supplementary material, would be useful to readers? The multiple overlapping dots on Fig. 1 do not provide any useful information in this regard.
• The Discussion compares the accuracy of the four methods you used, so the data on which this comparison is based must be clearly presented in the Results.
• Why did you use four methods and two use criteria (50%, 95%) and two sexes yet not mention any patterns in these sources of variation?
• What is the evidence for multiple core zones? Would this be apparent in figures for individual animal home ranges and data points?
• Was the tracking carried out for long enough to provide a stable home range estimate for each individual?
• There might be some interesting information to provide about the dispersing individual from a second study area, but it should not be simply mixed in with the others because it will distort the measures of central tendency.
• L183. Why do you regard P>0.01 as non-significant, instead of P>0.05, the normal convention? It is ok to have a different alpha, but you must explain and not use it arbitrarily.
• L187. It is not clear how the mean and variation were calculated. Was this based on the mean of the means of each individual or the mean of all data points? Providing sample size would make this clearer.
• The downscaling results in the supplementary figure do not correspond to the methods which imply only 2 intervals.
• L193. The first paragraph on habitat selection has no statistical support and was not clearly indicated in the Methods.
• L197. The second paragraph needs to explain how to interpret the figure and where the p-values come from. Consider a more logical organization of the variables.

Discussion
• The Discussion is not well organized. The order of topics should follow objectives, Methods and Results.
• For each topic, the Discussion should include a brief (1 or 2 sentences) review of what was found (i.e., key results), then a critical evaluation of the strengths and limitations of the findings (for example, was there anything in the methods, habitat, observation period, amount of data that could have resulted in an over- or underestimate?), then relate the findings quantitatively to the literature, including highlighting any novel or anomalous findings. In the present discussion of home range, a number of previous studies are cited, but it is not clear how your findings do or do not fit in with these previous studies.

References
The references need careful work. There are many errors and inconsistencies in the use of capital letters in article titles, lack of italics for species names, missing information, and additional incorrect words. In-line citations sometimes show incorrect information (for example, first names).

Table headings and figure captions
• The headings and captions are incomplete. All abbreviations need to be clearly identified. Units need to be identified.
• The supplementary figures also need complete captions.
• Figures and tables should be numbered in the order cited in the manuscript.
• Tables should not include vertical lines
• Figures should not rely simply on color because some readers are color blind or will be using copies printed in black and white.
• Fig. 4 lines are very small and hard to see. Look at some published examples to see how to improve the figure design. Is there any logic in the vertical order of variables?

Reviewer 1 ·

Basic reporting

Figures are not always clearly labelled. The raw data are not supplied and are not adequately described. Details in my additional comments to the authors.

Experimental design

The research question should be better defined. It would greatly help if the last paragraph of the introduction was formulated as a set of research objectives. The methods are not described with sufficient detail & information to allow replication. Details in my additional comments to the authors.

Validity of the findings

Some underlying data have not been provided. Not all conclusions are well supported by results. Details in my additional comments to the authors.

Additional comments

The authors report on the space use of mountain red foxes (Vulpes vulpes montana) studied with GPS collars in northern India. The study could have regional and taxonomic significance and could be a useful addition to the literature on red fox space use. Indeed, this generalist species is studied throughout the world to better understand the ecology of mammal meso-carnivores, there is interest in comparing the ecology of populations across habitats or taxonomic units, and very little is known of the studied subspecies.

Unfortunately, however, the paper suffers from many problems of form and content, including poorly focused objectives, absence of critical information about the dataset, drawbacks in the analyses, and need for improvement of figures and references.

I hope that my specific comments below can help the authors to improve the manuscript so it can be published.

Introduction
The last paragraph is critical and it should clearly outline the objectives of the study (and ideally some tested hypotheses, if appropriate). I suggest outlining a number of specific objectives, such as determining home range size and daily movement rate of Vulpes vulpes montana in a village of Ladakh in the North-western Trans-Himalayan region of India. I also suggest to select only a few objectives, and remove those with limited scientific interest, such as the directionality of movements (see below).

Methods
The described study area actually consists in two study areas separated by ca. 80 km. Six foxes were tracked in the western study area (Chiktan village) and one in the eastern study area (Hemis National Park). The fox tracked in Hemis National Park dispersed and actually used two separate home ranges. I suggest removing this individual from the paper as its data are very difficult to compare from those from Chiktan.
The dataset must be much better described. A table should indicate the body mass, sex, ID (F1, F2, M1, etc.), first tracking day, last tracking day, length of tracking period, and number of GPS fixes for each individual.
The raw data (GPS locations) must be provided. The best way to do this is to deposit the GPS locations in the public repository Movebank.org. In addition to allowing replication of the analyses, this might generate interesting future collaborations once other researchers know of the existence of this dataset. Data deposit in Movebank is free and people managing Movebank are very supportive and can help with technical details if needed.
L 116. I assume this is 100 mm per year. Please specify.
L. 117. Temperatures go down to -30C (rather than up to).
L. 138. “We chose 15 minutes time interval as it gives robust results for the actual daily distance travelled by animals (Musiani, Okarma & Jȩdrzejewski, 1998).” This is not exact, as shown by Poulin et al. 2021 cited later in the paper. 15 min underestimates traveled distances, especially when foxes are hunting. However, the data remain interesting even with 15 min intervals as this allows comparison with many other studies.
L. 152-157. I did not understand this paragraph about false positive errors. It seems that this paragraph refers to the filtering of the data and complements L. 144-146 but this is not clear to me.
Fig. 1. The legend is incomplete. What exactly are the green and cream areas on the small map of India? Also, the boundaries of Hemis National Park and Ladakh UT are not easy to see on the big map. What does UT mean in Ladakh UT? Village/city names on the large map are not readable. Symbols for F1, F2 and F3 are impossible to recognize on my pdf version of the figure (they all appear white).

Results
L. 168-169 are parts of Methods.
L. 170-171. As indicated above F3 is really different from all other individuals as it dispersed. It is not valid to calculate a home range for a dispersing individual.
L. 173-174. The core areas could be shown with a map zoomed on the Chiktan study area.
L. 179. I don’t think that the directionality of movements is very interesting for territorial animals, unless you are interested to show that they tend to follow valley bottoms. In that case the directionality should be associated with a clear objective/hypothesis. (I agree with the discussion L. 238-246)
L. 180. How do you know that dens were at the center of home ranges? What data support this?
L185-187. Fig. 2 seems to show movement rates calculated with 2 hr-intervals but this is not indicated on the figure. 2 hours is a major underestimation. The daily movement rates reported are thus only APPARENT movement rates and this should be specified.
L.192. F3 should be excluded from this analysis given that it lives in a very different habitat in addition to be a disperser with two different home ranges.
S.3 is not cited in the Results section. It is only cited in the discussion, after S.4 had been cited in Results.
Table S2. Some daily average movements (in m) are reported with 5 decimals, which does not fit at all the precision of the GPS or the biology of the species. It is more valid to report daily average movements in km, with two decimals (10-m precision).
Supplemental figures. I have found no legend for the four supplemental figures.
L. 192 (Habitat selection)
There is a difference between habitat preference, habitat use, and habitat selection, and words seem to be sometimes used interchangeably. In this study you should only refer to habitat use (to describe where foxes go) and habitat selection (to describe which habitats are more used than random). It is essential to present a habitat map.

Discussion
L. 210. “in our study we found LoCoH and BBMM were the best suitable methods for estimation of home range”. What is the justification for this statement?
L. 216-219. Yes, I agree about this caution to consider nomadic foxes while calculating home ranges. Therefore, you should not present home range calculations for this individual.
More generally, we do not know if observed home range sizes reached their asymptote (i.e. if enough GPS fixes were obtained for each individual so that their home range size calculations were meaningful), therefore it is impossible to interpret them correctly.
L. 260-263. “Although the actual distance travelled by an animal as compared to the straight line distances obtained from the GPS fixes may be an under-estimate, we also assume that additional errors pertaining to the accuracy of GPS fixes in the short 15 minute time interval may nullify this effect.” Unfortunately, there is no good reason to make this assumption. The underestimation due to the subsampling of true locations may be much higher than the overestimation due to GPS error.
L.263. “For example, despite being inside the den the GPS fixes may vary and show a continuous movement.” When animals are inside the den, no GPS fix is obtained because the collar cannot communicate with the satellites. However, the argument is valid when animals are resting outside the den, with a clear view of the sky.
L. 275. “smaller individuals”. As mentioned earlier, please give body sizes.
L. 282, “The presence of water bodies is also known to have a positive influence because of the presence of rodents and has been reported from previous studies (Szor, Berteaux & Gauthier, 2008”. This reference is about lemmings in the Arctic tundra; it would be useful to indicate why you think it is valid in your study area.
L. 304. “The radio-collar for F2 individual worked for lower number of days (56 days).” This is very important to know and explains why more details must be given in Methods about the dataset.
L. 326-327. “Although, we did not find any significant directionality in the movement of red fox this is the first study that explores this possibility.” Didn’t you find some directionality according to topography (valley orientation)?

References
More care is needed regarding reference formatting. For example:
L. 383. Title should not be in capital letters.
L. 439 and 444. Family names must come first.
L. 451 and L. 565. Reference is incomplete.

Reviewer 2 ·

Basic reporting

no comment

Experimental design

no comment

Validity of the findings

no comment

Additional comments

This manuscript describes results of a study of home range size, movements, and habitat use of 7 red foxes from the trans-Himalayan region of Ladakh, India. I agree with the authors that although red foxes are widely distributed and have been studied in many parts of their global range, there is exceedingly little information published on high-elevation populations, some of which are highly endangered. Consequently, I found the study to be of great interest.

I appreciate the authors’ efforts to interpret home range estimates critically and explicitly in the context of residence status, correctly distinguishing an individual who made dispersal movements
One of the interpretations in the Discussion for differences among individuals in agricultural habitat use was body size, which is an interesting thought, although it is unclear to me which direction was a more likely causal link (i.e., larger body size indicates social dominance and first choice of habitats, favoring ag habitat or access to resources in ag habitat facilitates larger body size). In either case, no data were presented on body size. I would request that this information be added to Table 1. It would also be useful to indicate which, if any, individuals shared a territory (e.g., family members or mates). There are so few individuals in the study that it would be valuable to provide as much data on those individuals as possible. Along the same lines, several supplementary figures and tables were referenced in the manuscript, yet I did not have access to any of these in my review.

Regarding presentation of results, I think Figure 1 could be improved considerably by showing insets zooming in on the two clusters of telemetry points so that it is apparent how individuals use space relative to one another. Also, the dots could be made larger without borders so that colors would be more easily discerned. As presented, it is impossible to differentiate the locations of the different individuals. Finally, each of the insets should have the habitat types so that readers can see how locations clustered relative to these habitat types, especially agriculture and human settlements. The figure should also be referenced first in the results given it includes the locations of foxes.

Otherwise, I have made several minor comments and suggested edits on the attached pdf.

Annotated reviews are not available for download in order to protect the identity of reviewers who chose to remain anonymous.

---

## Round 0.2 · Major Revisions

The reviewers and I agree that your manuscript has been substantially improved. However, there are numerous problems remaining, requiring another round of revision. The reviewers have provided a number of suggestions, and I have many more. Some of these problems are more evident now that the manuscript is clearer. For example, I found unexplained inconsistencies in reported data. Others are due to failing to respond completely to previous suggestions. For example, there are still mistakes in spelling and abbreviations, capitalization in references, and use of color in figures. Note that when authors do not carefully check their manuscripts, editors and reviewers have to repeat points they already made, creating some frustration due to additional work that should not be necessary. I urge you to be meticulous in checking that you make all the changes required (or explain why suggested changes are not appropriate).

General comments
Both reviewers note that the prediction of directionality of movement based on directionality of hunting is not logical and that the results do not support this empirically. I agree with them that this material should be removed from all sections of the manuscript.

Instead of finding a fluent English-speaker familiar with scientific writing in ecology to carefully read your manuscript, you used an online app to correct the language use. This provided some improvement but was clearly not adequate as you can see by the numerous errors I had to correct. I think that my suggestions will be adequate for the text that continues from this to the final version, but please seek help so that any new material avoids errors that would require yet another revision.

I requested that you check for consistency in use of SI units and U.K. spelling. Although you indicated that you had done so, this was not complete. Some examples are kms in Figs. 1, S2; mins and hr in Table S2; meters in Fig. S3, kilometers in Table S2. The SI abbreviation for day is d.

Note that when units with a number are used as an adjective, you need a hyphen. This is not the case when the units are used as a noun. For example, ‘we set the GPS to 15-min intervals’ but ‘our data were based on intervals of 15 min.’ Please check the entire text; I corrected some but not all cases.

There is a hyphen in p-value and t-test.

To indicate variability of a mean for descriptive data, SD is more appropriate than SE. Please replace SE with SD in the text and tables.

PeerJ Instructions of Authors ask that you do not justify (align) the right hand margin of the text. Please correct this.

My minor corrections are on an attached pdf. As last time, I used the pdf provided by Reviewer 2 so that you would have suggestions from both of us on the same document.

Specific Comments
Abstract
In addition to mean and SD, the range of home range areas and of daily movements would be useful.

Key Words: add Canidae

Introduction
The organization and flow of idea in the Introduction is much improved.
The orientation information should be removed.

Methods
The Methods are now clearer and more complete.
L118. Please confirm that the spelling of ‘North-western Trans-Himalayan’ is the correct official form; capitalization of North-Western would occur only as an official region, and northwestern is not usually hyphenated.
L125. If the religion is relevant, you should briefly explain why. Otherwise, delete. The sentence seems to imply that the abundance of anthropogenic food sources is related to the community being Muslim.
L122-127. Retain the information about the village and the park in this section but move the information about the number of foxes collared in each area down to the next section. This goes with the start of the section on Subjects (L134). You imply that the female in the park was attracted to habitation so you should probably indicate that people lived in the park and whether there were whole villages are just occasional houses.
L141. I do not recall that you do not report measurements except weight anywhere. If you have no need to report them, you can delete the mention in the Methods. Otherwise, tell what measurements you made and how you made them.
L141. Somewhere in this area you should give an overview of the durations of tracking (provided in specific detail in Table 1), for example the range of days. You should also add a comment about the reasons for the variable durations. Were these due to battery failure, premature collar release, death of the individual or some other reason? As I requested in my previous comments, it would be helpful if you provided some indication of the phenological context of the study period. For example, if this was a temperate area, this would be in the fall and winter, extending perhaps into Spring. How does this relate to the breeding season? It would be useful for readers to have an idea whether females might be feeding pups, whether males might be making longer trips for mating. There are a few relevant comments in the Discussion, but readers should know this information much earlier in the manuscript.
L155. Advantages of different Method should be move from Discussion to Methods here.
L163. Do you mean only two errors in the whole data set or two types of errors?
L168. Indicate why you wanted to know the effect of downsampling? It seems obvious that the distance will be shorter with fewer records? Did you want to quantify the effect or have some other reason?
L179-180, 197-198. These terms need to be defined and the units provided. For example, it is not obvious what the measure of ruggedness is or how much or what kind of vegetation it takes to change the category from barren to vegetation. On L241, when we find that all individuals avoided slopes, we don’t know if this means that they avoided all slopes or that steepness was associated with greater avoidance.
L180. Is water an anthropogenic disturbance? If not, revise the sentence.
L202. I did not see a clear explanation for how you calculated statistical significance for preference for landscape and topographic features.

Results
I agree with Reviewer 2 that it would be useful to provide additional information about the individuals, if known. For example, were they all mature individuals? Were they breeding? Did they have pups? Were they mates? You mention dominance, but it is not clear if you have behavioral observations or only inferred this from weight differences. You mention breeding females only late in the Discussion. This information could go at the start of the Results section if you gained information by following their movements but is appropriate to Methods if you the information at the time of collaring.

Home ranges. The text here is not very helpful, partially overlapping with Table 2 but not providing any synthesis.
• For example, you could point out means and ranges among individuals.
• You could point out whether the methods provide consistent values or whether there are consistent differences.
• The MCP core average is strongly affected by an extreme outlier is many times larger than the estimate by any other methods. Could this be an error or could it be an artifact of the method? From Fig. S2, it doesn’t seem to be that different.
• Similarly, MCP 95% average is strongly affected by the extreme value of M1 which is many times greater than estimates from the other methods. From Fig.1a, it looks like M1 HR might be about three times as large as M3, but not more than 20 times as large.
• The logical basis of reporting BBMM in the text and MCP in Fig. 1 is not clear.
• You should provide an overview of the results arising from the different methods of estimation (and discuss the appropriateness of different methods in relation to what are used by researchers in the Discussion).
• It seems to me that the great overlap of home ranges indicated in Fig. 1 is worth reporting (and later discussing whether this is normal for foxes). Do the core areas similarly overlap? Are there pairs as defined by overlapping cores? Do you know where the dens are in relation to the cores? Are the cores related to denning sites or feeding sites or something else?
• This is the section where you would note the individuals that have multiple cores and how convincing the evidence is. For F3, does the diagram take into account the move?
• You need to be careful to focus on home range here because daily movement belongs in the next section. The question here is whether F3 had an established home range. If there are three, you need to guide the reader to see the basis on which you drew this conclusion.
• You later refer to a relationship between weight and home range. If you want to refer to it in the discussion, you should point it out here. I could not see this immediately, so I plotted BBMM core and 95% data and still found it hard to see a link. With your small sample size, you won’t have statistically significant correlations, but could point out if there are indications of a trend (and if this is known from other studies). You must be clear about the evidence, not just state the pattern that you perceive.
• Similarly, if there is any trend toward a different between sexes, this would be the place to note them. I couldn’t see any trend when I plotted the data.
• In the previous version, you were asked about stable home ranges. You quoted the sample sizes used by other researchers. However, you animals might not be the same. Could you plot home range vs number of fixes to see if it reaches an asymptote? This is important for the reliability of your findings.
• Is F3 dispersing or traveling long distances with one or more home ranges? Fig. S3 is not completely clear because it refers to displacement. In the methods, you wrote that distance travelled is the sum of displacements. However, the text and caption implies this is displacement from the collaring location. Your Methods should prepare the reader for analyses that relate to displacements in relation to the collaring location as well as displacements between successive fixes. If Fig. S3 is displacement from the collaring location, it looks to me as though F3 remained stably located about 11 km from the collaring point, perhaps moving after being collared. Then after about 2400 fixes she moved to another location about 28 km away and then another location 32 km away before moving back to the 28 km location. Is this correct? Can you shed any light on this? When you calculated home range, did you take into account these home range relocations? It would not be valid to treat them as if they were a single home range. Ideally, you would calculate home range for the different time periods or at least indicate the problem. Simply stating that it was dispersing so you didn’t include its average is not sufficient, especially since it was not continuously changing central location but rather shifting between central locations.
Distance travelled per day
• As with the previous sub-section, the Results should provide a bit more context here. You only have one sentence talking about your main result, even though your Introduction and Discussion indicate how important it is.
• The text mentions Fig. 2 and Table S2 but doesn’t mention Table 2 which also has means.
• There are substantially discrepancies between the means in Table 2 and the text on the one hand and Table S2 on the other. For example, M4 is the lowest in Table 2 and one of the highest in Table S2. Neither seems completely consistent with Fig. 2. Either there is an important error here or the data has not been provided to understand the differences.
• The distribution of distances travelled and whether they change over time are potentially important: some days with no travel, many days with little travel and a few with long distances. Is this correct? The discussion should address the significance of this.
• Fig. 2 shows that the data are not normally distributed, so the mean and SD are not valid measures of central tendency and variability. Non-parametric measures are required (median, interquartile range).
• The caption to Fig. 2 is incomplete. I guess these are violin plots. If so, I understand the distance of the edges of the plot from the middle line, but I don’t understand the horizontal distribution of the individual data points.
• If I am interpreting it correctly, Fig. S2 suggests that all individuals except F3 travel less than 5 km on most days whereas F3 travelled about 11 km/d until after about 2400 fixes, she started to travel about 30 km/d. These values do not agree with Table 2. I may have misunderstood something. If I have misunderstood, other readers will so you have to explain. There are possible biological and statistical issues to be understood here.
• Effect of downsampling. You never said why you carried out the downsampling. It seems obvious that downsampling would reduce estimated distance travel. Is there an implied quantitative question such as how much is estimated distance reduced by reducing fixes by half? Your objectives should indicate why this was done and your results should present the relevant information.
Habitat selection
• L230-234. This section does not agree with the synthesis provided in the Discussion on L210-213. Here in the Results, it is important to help the readers to see any patterns you found. You need to indicate not only the figure number but also what in the figure leads you to draw these conclusions. My understanding of your approach to habitat selection was to compare actual steps to random steps. However, you seem to be emphasizing here the difference in relative height of the bars for both actual and random in some landscape features is compared to others. It seems to me that it is relevant to help the reader understand both findings.

Discussion
Home range
L246-254 relate to Methods, not Discussion.
L254-256 are Results, not Discussion.
The Discussion of home ranges should start with a brief reminder of how home range estimates varied (or not) among different estimation methods and a clear and justified decision as to which is the best method to compare to the literature. (It is not enough to just give the reported advantages of each without relating them to your own study.) Then give an overview of your findings using the most appropriate method(s), including any possible biases such as whether the sampling duration was sufficient for a valid measure or whether home-range relocation occurred during the study. Then relate your measures specifically to the literature. You indicate that your population and habitat are poorly studied, so it is important that you make a clear statement about what contribution your study makes.
L273. There must be some speculation in the literature regarding why foxes vary so much in home range within a population. How do your data relate to these suggestions?
L273. Should you discuss possible roles and importance of highly overlapping home ranges? These individuals can’t be territorial, at least with each other. If you trapped in a single location, is it likely that you ended up with foxes that shared a territory or home range?
L275. I disagree with the reviewer that you should eliminate the discussion of F3. However, you need to work on how you discuss this individual. I did not see evidence in Fig. S3 of continuous dispersal but I could see the possibility of two or three different home ranges. This needs to be brought out in the Results with clear description of what in Fig. S3 constitutes your evidence and how strong it is. Here, you should discuss what you think is happening. Isn’t it possible that F3 was on an extended trip outside of its home range when it discovered your bait and was trapped rather than that it moved after trapping?
Movement
L381ff. You need to reorganize and develop this entire sub-section.
• Start with a brief overview of what you found, average and range.
• Then address the reliability of your measures. This is where the discussion of your downsampling results could be provided.
• Then, discuss potential correlates of variation, including the relationship of home range size to distance traveled, sex, and other relevant variables in your study.
• Then, compare your results to the previously published study, pointing out similarities and differences and offering explanations for these. For example, what was the habitat of the foxes in the previous study? Did their method possibly over- or under-estimate distance? Are distances similar despite differences in habitat? This discussion could include the correlates of variation in distance moved in other studies of this species or other similar species.
• Your topic sentence about movement as a key factor is a rather weak general statement and not appropriate for the first sentence. It would be much more useful for your discussion to show the implications of your findings on distance for understanding home range and territory. That is, don’t just say it, but show it with your own results.
• L290. The statistical significance of the reduction in estimated distance with downsampling is not the relevant point, since everyone would expect that. The amount of reduction might be relevant and potentially compared to other studies. This is where you would address the implications for why you did this in the first place.
Habitat selection
• L210-213. Here you present interesting material on the habitat use of individuals of different weight. However, your argument is too brief and unclear. Although the patterns described here do seem generally in agreement with the figure, they do not agree with how you presented the Results. (I have made some notes on this above.)
• Furthermore, you describe F1 as larger, but she is the second smallest subject. If you consider her small because she is smaller than the other female even though she is smaller than the males, you need to present information on how any size sexual dimorphism should be taken into account. You need an argument why you think a 400 g difference in weight (only 7% less) is sufficient to make a difference in dominance.
• You never described what ‘barren’ land is so we don’t know what high use of barren land might imply. We need some explanation of why a fox would be spending so much time on barren land and snow. If animals were foraging around habitations, what category would they be recorded in?
• You indicate that the animals using more agricultural habitat are dominant, but you do not say how you know this. It seems that the great overlap of home ranges is relevant here. Do foxes with highly overlapping home ranges nevertheless exclude others from whole habitat categories? It seems a bit unlikely to me, so you need to explain what might be going on.
• Your Discussion seems to relate only to the relative amount of habitat, not to the step choices. Since you argued for the benefits of this method, you should discuss what you found.
• You should present your results first, then relate to the literature rather than the other way around as at present.
Limitations of the study
• I do not think you need this section. The material is interesting and relevant, but is normal discussion and should be included as part of your discussion of the reliability of the data in each of the appropriate sections above. It is important that readers understand the data reliability when they are assessing your conclusions not at the end of the article.
• L370ff. I agree with the reviewer that this paragraph is too vague and generic. It includes very little that you discovered from your study and basically restates the justification rather than highlighting anything new. You keep repeating the importance of home range and travel distances. This could be replaced by a stronger Conclusions section in which you do not repeat your findings but instead point out the specific implications of your actual findings for the actual conservation of this species in this habitat.

References
There are still many mistakes in the references. I noted some, but it is your job, not mine, to find them all and correct them.

Figures and Tables
Fig. 1. This figure is much improved. It still depends on color, but it is hard to think of an alternative for the home range boundaries. However, you could add the identity of each individual in small letters in a non-overlapping part of each boundary line for additional clarity. All panels need to have letters and be described in the caption. Check instructions to authors; I think PeerJ prefers capital letters for panels. Units for elevation? Incorrect abbreviation for km. Boundary of park too thin to be seen.
Fig. 2. Caption must explain the plot fully (points, curved lines, bar, horizontal line, etc.). y-axis label should be Distance travelled (km/d). No need for color to separate sexes; figure can be in black-and-white; caption should indicate meaning of F and M; remove key. Space between 0-line and sample size should be consistent.
Fig. 3. Because the key extends the horizontal length of the figure and the category labels extend the vertical height of the figure, the actual information will be of reduced size when the figure appears on the page; this can be corrected by small changes. Use letters to indicate category and describe in caption, for example: ‘agriculture (A), barren (B), . . . Remove the key and describe in the caption. This figure is not color-blind friendly. You could add hatch marks to one category of bar or use grey and white bars. The axis labels and numbers are too small to be easily read. Keep the same vertical scale on all panels for easier visual comparison. Caption should indicate what the bars represent (proportion of what?). Caption should indicate what kind of analysis provided these measures and clearly indicate what actual and random mean.
Fig. 4. Caption is far too brief. You must explain everything in the figure so a reader can easily understand (dots, horizontal lines, vertical lines, etc.). There is no label on the x-axis and too much space between the categories on the y-axis. Not color-blind friendly.
Table 1. Same precision on all weight measures.
Table 2. Incomplete heading. SD not SE. Consistent precision within each column. Define all abbreviations. Some outliers need to be checked. If in error, recalculation needed which will affect the manuscript. If not in error, these large discrepancies should be pointed out and explained in Results and/or Discussion or both, as appropriate.
Supplementary figures
S1. Caption is incomplete. Reader cannot easily see the data. Does the x-axis need to be so long; I can’t see data points beyond about 14? Unclear label on y-axis. Can it be switched to proportion? Not SI units.
S2. Caption incomplete. Labels of panels and axes too small to be read. If scale differs among panels, it should be noted in caption.
S3. Caption very incomplete. Not UK spelling on y-axis. y-axis label should be Displacement from collaring location (m). Why not use km as in other figures and avoid wide axis label? Wasted space above 6,000 fixes will reduce size of final figure. ‘Linear’ is unexplained and may be described in caption and removed from panel. Any statistical method applied here needs to be in Methods.
S4 will probably be removed. If not, see comments for other figures to check for needed improvements.
Supplementary tables
S1 will probably be removed. If not, complete heading, remove box, fix spelling (static, p-value), use consistent precision.
S2. Check recent PeerJ articles for proper formatting of tables. t-test, p-value, space between number and units, SI abbreviations (mins, hr), more complete heading, p-value does not need precision more than three decimal places.

Reviewer 1 ·

Basic reporting

Clear, unambiguous, professional English language used throughout: Fair

Intro & background to show context: Yes

Literature well referenced & relevant: Yes with the exception of the literature on movement direction (there is some confusion with the literature on body orientation).

Structure conforms to PeerJ standards, discipline norm, or improved for clarity: Yes

Figures are relevant, high quality, well labelled & described: It's ok.

Raw data supplied (see PeerJ policy) : Yes

Experimental design

Original primary research within Scope of the journal: yes

Research question well defined, relevant & meaningful. It is stated how the research fills an identified knowledge gap: yes but one hypothesis is invalid

Rigorous investigation performed to a high technical & ethical standard: fair

Methods described with sufficient detail & information to replicate: yes

Validity of the findings

(Impact and novelty not assessed.)
Meaningful replication encouraged where rationale & benefit to literature is clearly stated:

All underlying data have been provided; they are robust, statistically sound, & controlled: yes

Conclusions are well stated, linked to original research question & limited to supporting results: yes

Additional comments

The revised paper is clearly improved over the first version. I still do have some critical comments below but I trust that they can be fixed by the authors.

Introduction

L 107-109. The formulation of the hypothesis about directionality of movement is not valid. Cerveny et al. (2011) did show that the body direction of foxes as they attack their prey was non-random (foxes tended to direct their jumps in a roughly north-eastern compass direction), but your data do not describe the body orientation of foxes, but rather the direction of their movements. These are two very different things. In an animal with a rather circular home range, you should anticipate no directionality of movements, while in an animal with a rather elongated home range, you should anticipate some directionality of movements (according to the longest dimension of the home range).

L. 301-303 of the discussion touch on this, and confirm that the formulation of the hypothesis was not valid.

Methods

L. 150. You indicate that the data from seven red foxes included 31,261 GPS fixes. I suppose that this is before filtering? The Excel file contains 28,163 GPS fixes. I suppose that this is after filtering? It would be useful to make this clear in the text of the paper (perhaps indicate the number of fixes after filtering at the beginning of the Results section).

L. 151. You indicate that “The filtered data excluded fox locations that exceeded > 48 km/h as this is the highest speed of the red fox (Haltenorth, & Roth, 1968)”. But then in the discussion you indicate “Hence, our method of restraining the maximum speed limit of 9.35 km/h for red fox may be a fair estimate.” I do not understand the discrepancy between these two sentences. It is possible that I missed something during my reading.

Results

L. 236-24. There are still format problems in this paragraph and in quite a few other places. Here please check P<0.001 on lines 237 and 242.

Table S.2. P values should not include that many decimals. Four (as in Table S.1) is enough.

Discussion

L. 259. Lai et al. (Journal of Mammalogy 2022, https://doi.org/10.1093/jmammal/gyab164) have recently reported red fox home ranges of 60 to 73 km2 in the Arctic.

Reviewer 2 ·

Basic reporting

I think this draft is much improved. I made several suggestions on a pdf of this ms for deletions that I think would simplify the manuscript. I think that the language usage is much better and could be improved still. However, most of the text was clear enough to me.

Experimental design

see previous review

Validity of the findings

see previous review

Annotated reviews are not available for download in order to protect the identity of reviewers who chose to remain anonymous.

---

## Round 0.3 · Minor Revisions

Reviewer 1 was not available to check this revision. I therefore sent the manuscript to a third reviewer who provided a detailed and useful review along with that of Reviewer 2. Both reviewers indicate that the manuscript continues to improve but that it needs additional work before it can be published. Unfortunately, some of the remaining errors include problems that were identified in previous versions and still have not been completely addressed (for example, references, use of SI units). This is very frustrating to editors. I have provided more detailed suggestions that usual because I would like to see the next version as acceptable for publication. However, I will not hesitate to return the manuscript again if errors persist, including the numerous problems with English usage.

I apologize for the delay in completing my review. I had some health issues that reduced the time I could devote to the manuscript. However, you should also note that persistent problems with the manuscript (both scientific content and presentation organization and style) meant that much more time than usual was required to complete my editorial review.

I am pleased that the third reviewer raised the issue of author responsibilities. I note that the two authors who are identified as project supervisors, one of whom is the corresponding author, are also experienced scientists with numerous publications to their credit. If these authors did not see the manuscript before resubmission, there is a problem because all authors must approve of the submission. If these authors did see the manuscript but allowed resubmission with so many remaining errors, there is also a problem. The problem is that these authors are exploiting the volunteer time of the reviewers and editor to do the work that they should do before submission or re-submission. This is unfair to all engaged in this voluntary effort, and it is not conducive to efficient publishing of strong science.

To help guide your response to the reviews, I provide additional comments below on some of the concerns that they raised. Reviewer 2 referred to line numbers from the Track Changes Word document; these are sometimes inconsistent depending on the format of the Word file on a particular computer, so I have used the pdf line numbers in my comments. In addition, I checked all the grammatical suggestions and incorporated them with my own suggestions on the pdf. I have provided much more detailed feedback and suggestions for wording in the hope that we can complete this project. However, that depends on thorough and careful revision by the authors.

Editor's Notes and Comments

1) Location Errors (L145-150). Both reviewers indicated that this section was confusing. Referring to ‘error types’ and ‘false positive’ could lead a reader to think that these are somehow associated with type 1 and type 2 errors in a statistical test, so you need an alternate wording. It is also not clear if you identified these errors only using the speed of displacement or from direct observations or by some other approach. As Reviewer 2 points out, there are other types of possible error that are not mentioned, and it is not clear why. The relationship between the 48 km/h criterion and the 9.35 km/h (L150) is also not clearly explained. Did you apply the 9.35 km/h threshold to all data or only some? How did you arrive at the 9.35 km/h threshold. Your rebuttal letter is somewhat clearer, though still incomplete, but it is very important that you explain what you did in the manuscript itself, not just in a response to reviewers/editor. Please revise this section clearly and concisely and consult with colleagues to be certain that it communicates a clear and accurate statement of what you did.

2) You use different terms, 'land use' and 'land cover' and sometimes an awkward phrase including both terms to refer to the same thing. Reviewer 3 suggests 'land cover' as the most appropriate term, but you referred to land use more frequently. I don't think the term is usually hyphenated; please check for correct spelling in a reliable source (for example a major established journal). Please decide on the most appropriate term and its correct spelling and use it consistently throughout the manuscript (including figures and tables!).

3) Although Reviewer 1 suggested removing the repetition of 10 random steps on L171,177 and the repetition of land cover categories L174, 205, I think is acceptable to leave them in the manuscript if they help make the revised text clearer.

4) I agree with Rev. 3 about relocating information about the step function and exclusion of F3 from home range estimates and have provided suggestions for alternative text on the pdf.

5) I agree with Reviewer 3 that the overlap of home ranges should be stated in the Results. You do not need to remove it from the Discussion because that is where you discuss possible reasons for this overlap.

6) Rev. 3 provides two additional references that seem very relevant to your study. Please integrate them carefully into your manuscript. One of the references may require you to modify one statement about the novelty of your contribution and add to the comparative discussion of your movement data.

7) Rev. 3 suggests that there should be more discussion of the results of the downsampling. Previous reviewers and I have tried to get you to address this in previous versions but you seem unwilling or unable to do so. I will therefore reluctantly accept this part of the Discussion, including the reference to the danger of high sampling rates, without requesting more changes.

8) Rev. 3 suggests that it would be better to separate days according to the photoperiod instead of using midnight. This is a sensible suggestion. However, this point was not raised by previous reviewers (probably because you did not explain what how you did it until I requested the information). At this stage of the manuscript development, I feel it would be very disruptive to make the change and suggest that you leave the separation of days at midnight as it is. The reviewer notes that a change, while more logical, is unlikely to make a substantial change to the results.

9) Dominance. L345 refers to dominance without any prior mention or direct evidence. I have provided a revised sentence that I believe captures the argument you are trying to make but seems clearer. The meaning is not exactly the same as what you wrote, so you should not accept it unless you agree. If I have not expressed what you mean, please provide a revised version that makes it clear that the arguments using dominance are based on an assumption that this is associated with weight.

10) Habitat selection/use measures (L160ff). There is still some lack of clarity in this section. The paragraph starts with reference to habitat selection analysis based on applying SSF to several topographic, environmental and anthropogenic variables. On L172 land cover is introduced but it is not clear how this relates to the previously described SSF. Is it part of the same analysis or a different one? What question is being asked. Is the question different but the SSF the same? Then, on L177 the original variables (topography, etc.) are repeated. I tried to understand what you had done by turning to the Results. However, this was confusing too. Even though you should keep methods and results in the same order for ease of understanding, the results starts with the land cover and then goes to topography, implying that these are separate analyses. The way the data are presented in Fig. 3 and 4 is very different too, with statistical information only for the topographic variables. You must clarify and coordinate the methods and results on this topic, making it clear to readers what question you are asking, how you are approaching the question and what type of answers you can get. Keep subjects in the same order in all sections. I looked at your response to my query on the previous version and found the answer unclear. I do not understand why the land cover does not have the same statistical approach if the same method was used. This is a need revision where errors could accumulate. I hope you will be very careful that your revision is complete and clear and does not require additional revision.

11) Displacement distance. Reviewer 3 notes that the procedure for generating Fig. S3 was not mentioned in the Methods. There should be a clear statement of what you did and why. At present, this information is provided only in the Discussion. I have suggested revised wording and where it could go in the Results. For the Methods, I think a statement would be appropriate around L150, at the end of the daily movement methods and before the downsampling methods.

12) Daily movement. Reviewer 3 notes that you should only use the data from 15-min intervals to calculate daily movement. The Methods indicates that for M3 and M4 you used only the first 30 days for your downsampling analysis (L157-158). However, you do not specify that you restricted the data to this period for the movement analysis. In fact, the opposite is implied by L319. This is important for the reliability of your data. If you did include data with longer sampling intervals, you will need to recalculate these values, being sure that you have corrected all mentions in Results and Discussion, as well as adding this information to the Methods.

13) References. Despite your assurances, errors remain in the references. These are mostly minor issues of failure to capitalize proper nouns and capitalization of words in article titles, but there is also a published 2021 article listed as 'in press' without page numbers, and an incorrect journal title that amalgamates two different journals.

14) I have checked all the grammar, spelling and stylistic suggestions of both reviewers and incorporated them into the suggestions on my pdf. One comment by Rev. 3 refers to L140 but is a typo as the material referred to is on L240.
15) Fig. S1.
• Non-SI units on x-axis
• Uninformative label of y-axis
• Impossible to read all data. No points are visible above about 14, but the x-axis extends to 45 km/h (perhaps converting data to proportions and using a log scale on the y-axis would help; alternatively, separate figures for < 1 km/h and greater than or equal to 1 km/h might be clearer).
• Caption is incomplete, uses non-SI units. Give total sample size and provide an empirical description for 'distribution', e.g., proportion or total number of observations.
16) Fig. S2. Caption is incomplete. Does not identify abbreviation for home range estimator or abbreviations for individuals or the coding for probability of use. It should alert readers to the different spatial scales of different figures and indicate the program used to detect and plot these patterns.
17) Fig. S3.
• Caption incomplete: missing meaning and units of displacement, clarity that the x-axis represents successive GPS fixes so it provides a time line post collaring, description of the trend line and how it was calculated, specific mention of the abbreviations of individual, note that some individuals had longer intervals for some fixes, so the time scale will not be the same for all.
• Trend line should not extend beyond the data points for each individual
• If I read the figure correctly, F3 was 10 km from the collaring location at the first recording. Is this possible? Does it imply that recording did not start right after collaring. I do not recall a statement to this effect in the Methods. If it is correct that the animal moved this far before the first recording, a brief description or explanation in the caption may be required to help readers understand.

Additional comments
L110. Here and elsewhere: no period after Table (it is not an abbreviation)
L150. Here and elsewhere: no period after S in supplementary figures and tables
L152. As noted by the reviewer, downsampling does not alter the movement but the estimated movement; please search carefully to correct the wording in all instances.
L153. Careless use of non-SI units despite previous warning; in this case, spelling out is more appropriate anyway.
L173. 'Land-use land-cover 'as one expression is awkward. Decide which is the best term and use it consistently as noted above.
L174. As previously requested, it is important to define each of these categories at first use. For example, barren land probably contains some vegetation, so what is the distinction between these? Does snow refer only to permanent snow fields? Presumably, the source provides concise operational definitions that you can convey to the reader.
L192. The statement that BBMM and LoCoH produce smaller estimated home ranges than MCP and kernel method does not agree with the estimate for kernel which is very close to BBMM. Is there an error in the data presentation or does the sentence need revision?
L322. Explain why you think movements may have decreased after you were unable to collect data. If there is no logical reason, it should not be mentioned.
Table 1 does not need a box around it or lines separating each individual. Lines under the column heads are sufficient. Specify (day/month/year) for Date of collaring.

Reviewer 1 ·

Basic reporting

Line numbers refer to the file peerj-69037-Track_changes_on_second_revision.docx
The English and format are improved with each revision, but some problems persist or some new problems are introduced.
Examples:
- Many double spaces (e.g., L. 33, 346, 433, 435)
- L. 49 ladscape vs landscape
- L. 72 Word missing (space use)
- L. 162, 332, 345, etc. Remove the dot at Table.
Literature references and background ok
Structure of article and quality of figures and tables ok, but resolution of Fig. 2 appears low.
The submission represents an appropriate "unit of publication".

Experimental design

The research question and structure of the paper (1-home range, 2-daily movement, 3-habitat selection) is now clear.
Areas for improvement:
- L. 294-299. The section on false-positive errors introduces some confusion. How did you detect these errors? Through visual observation of animals or though inspection of the spatial data? Why are there only “two types” of errors? For example, did it happen that a fox was in an area of its home range that was neither the den nor a home range boundary, and yet the GPS fix indicated another location? Would this be an error 3? It would be sufficient to indicate that you inspected the speed data, identified some suspect spikes, and decided to remove data points containing speeds over a given threshold (if this is what you did). The concept of false-positive errors and the classification of errors into error 1 and error 2 types does not seem justified or useful.
- L. 300. “Since GPS fixes obtained from larger time intervals can affect the daily movement of animals”: Actually the time interval does not affect the daily movements. It only affects your ASSESSMENT of the daily movements.
- L 320 versus L 326. No need to repeat the information about the number of random steps per used step.
- L. 354. No need to repeat the information from L. 323. And the reference should go to the Method section.
- L 356-357. You appear to introduce some information about habitat selection in a paragraph devoted to habitat use. Please keep all information about habitat use in the first paragraph (L. 354-361) and the information about habitat selection in the second paragraph (L. 363-369)

Validity of the findings

The findings are valid. They are limited by small sample sizes but this is well acknowledged.
All underlying data have been provided; they are robust and statistically sound.
Conclusions are well stated, linked to original research question, and limited to supporting results.

Additional comments

This second revision improved again the manuscript although I still do have some critical comments, stated above.

·

Basic reporting

This version of the manuscript has already been revised accoring to the suggestions of reviewers and an editor. As I understand, the previous comments also concerned the language, which was improved. The present version is clear and reads fluently (I am not an native english speaker). However, I have a few more language comments:
Line 91: replace “comprise of“ with “belong to”
Line 103: … in Hemis National Park
Line 121: ... in winter
Line 125: … GPS fixes and were filtered…
line 173: decide whether you want to use the term land-use or land-cover and use one of them consistently. In my opinion land-cover would fit better to the categories you determined.
Line 183: Start maybe rather with “Our red foxes”
Line 140: inferred to here reads a bit strange. Maybe rather attributed to or explained by?
Line 269: This last sentence could be edited. Use maybe suggest instead of speculate or reformulate the whole to something like: In the case of our foxes the two factors could together explain the observed high variation in home range sizes.
Legend to Fig. 2: remove "per day"

The background of the study is rather well presented and relevant literature is cited. Statements about absolute novelty of such research in a wide-spread animal like red fox are however difficult. I suggest removing the sentence on line 79. Daily movement rates were for instance reported by
Jean-Steve Meia and Jean-Marc Weber. Home ranges and movements of red foxes in central Europe: stability despite environmental changes. Canadian Journal of Zoology. 73(10): 1960-1966. https://doi.org/10.1139/z95-230

Another reference which would be relevant to your study is
Main, MT, Davis, RA, Blake, D, Mills, H, Doherty, TS. Human impact overrides bioclimatic drivers of red fox home range size globally. Divers Distrib. 2020; 26: 1083– 1092. https://doi.org/10.1111/ddi.13115

Concerning the structure of the article, some information provided in the discussion could be moved to the methods or the results. The information of why you carry out the step selection function analysis at the individual level and not at the population level should be mentioned in the methods. The reason for not estimating a home range size for F3 should be mentioned in the results – or at least as a footnote to table 2. The reference to Poulin et al. (2021) regarding the impact of sampling frequency on estimates of daily movement rates could be given already on line 154. The fact that there was a large degree of overlap between the home ranges of the foxes is also a result and could be presented in the same paragraph as the home range sizes.
In general, the manuscript presents relevant results to the quite open question asked.

Experimental design

Experimental design
The manuscript describes original primary research and new data. The research question is well defined and aims at providing new information from a little studied area of the world. Consider however tuning down the novelty of daily movement rate estimates as mentioned above.
The methods are well described and adequate for the aims of the study. I have, however, a few questions:
Are you sure the Litetrack 150 collars were from Sirtrack? At present they are sold by Lotek.
It is interesting that four different estimates for the home range were calculated. However, I would have expected more of the comparison between the different estimated to be presented in the results, notably regarding the variation in size differences between individuals and methods. It seems the authors decided quite early that BBMM is the most reliable method and therefore mainly report that in the results. If this is so, I suggest to mention this first in the methods. I see this paragraph was already addressed in the previous review, but in my opinion it could still be improved a little.
Did you consider choosing a more biologically relevant day for the daily movement rate than from 0:00 to 24:00? Foxes are often nocturnal, and cutting the day in the middle of the night may not represent their activity pattern. By using astronomical tables you can determine the onset of the day, and for instance use from morning to morning or from evening to evening. But I am not sure that would really change your results.
The term “false positives” is a bit surprising when you mean GPS fix errors. False positives usually refer to some kind of tests. It is also not so clear why you chose exactly 9.35 km/h as cut off. Also, what is the unit of the y-axis on Fig. S1? And how did you determine what the fox was doing when Error 1 and error 2 occurred? If this is based on visual observations you should mention this, and also how you made sure you were watching the right fox.
For the estimation of daily movement rates, as you document in this study that the GPS interval is important, only the period with 15 min intervals should be used for all individuals.
Considering land-cover: snow depends on the season. What date did you use to estimate snow cover, or is this only the area of permanent snow fields?
Fig. S3: it is not clear why distance from the capture location is presented on this figure. This is not something that is addressed in the manuscript, although it is interesting that F3 seems to have dispersed.
Also many questions to the methods and the data were addressed by the previous review, and it is likely that the manuscript substantially improved for this version, but in my opinion not all of the have been fully solved

Validity of the findings

All the positions of the foxes are provided in an excel file.
For F3 some positions from 29 and 30 of September are included which were probably from when the collar was tested before it was deployed. These could be removed. I did not check anything else than the dates with these positions.

Regarding the discussion and the interpretation of the results, I thought that it was interesting that the two females (F1 and F2) that were the foxes not attracted to the settlement were also the foxes not avoiding roads and slopes. Could they maybe exploit anthropogenic resources along the road instead of close to the settlement? Also is it so that slopes are steaper along the road? Given that they don't avoid roads, I am not so convinced about the explanation that they avoided human disturbance (Line 370).

Another comment to the discussion regards dominance. How was dominance determined? You use the word for the first time in the discussion and it is not explained how you inferred it. If it is only based on the assumption that larger foxes are dominant, I suggest to remove it. Or maybe write in that paragraph something like "it is possible that larger foxes are dominant" with a reference. However, the present study is not about that, so these are only speculations.

Additional comments

Both in the introduction and in the conclusions conservation is mentioned in quite general terms. As red foxes are a species that is increasing in many parts of the world among others because of their ability to effectively use anthropogenic subsidies, it would be interesting to know what their status is in the study area. Are they increasing? Hunted? Are their concerns that they would have negative impacts on threatened prey species? Or are red foxes rare in the area?

Lastly I have a comment to the senior author(s) of this manuscript: The previous review and the replies reflect that a lot of effort has been made by the reviewers (and the authors) to improve the manuscript, including the language. However, there are still things to improve. As co-authors, I see some of this work as your responsibility of supervising your younger or less experienced colleagues. This is of course my personal opinion.

---

## Round 0.4 · accepted · Accept

Thank you for your careful attention to the needed changes. I now consider the manuscript ready for publication.